# Error-Corrected Deep Targeted Sequencing of Circulating Cell-Free DNA from Colorectal Cancer Patients for Sensitive Detection of Circulating Tumor DNA

**DOI:** 10.3390/ijms25084252

**Published:** 2024-04-11

**Authors:** Amanda Frydendahl, Mads Heilskov Rasmussen, Sarah Østrup Jensen, Tenna Vesterman Henriksen, Christina Demuth, Mathilde Diekema, Henrik Jørn Ditzel, Sara Witting Christensen Wen, Jakob Skou Pedersen, Lars Dyrskjøt, Claus Lindbjerg Andersen

**Affiliations:** 1Department of Molecular Medicine, Aarhus University Hospital, 8200 Aarhus, Denmark; amandafbj@clin.au.dk (A.F.); sarah.jensen@cruk.cam.ac.uk (S.Ø.J.); tvh@clin.au.dk (T.V.H.); demuth@clin.au.dk (C.D.); mathildediekema@clin.au.dk (M.D.); jakob.skou@clin.au.dk (J.S.P.); lars@clin.au.dk (L.D.); 2Department of Clinical Medicine, Aarhus University, 8000 Aarhus, Denmark; 3Institute of Molecular Medicine, University of Southern Denmark, 5000 Odense, Denmark; hditzel@health.sdu.dk; 4Department of Oncology, Odense University Hospital, 5000 Odense, Denmark; 5Department of Oncology, Lillebaelt Hospital, 7100 Vejle, Denmark; sara.witting.christensen.wen@rsyd.dk; 6Bioinformatics Research Center, Faculty of Science, Aarhus University, 8000 Aarhus, Denmark

**Keywords:** ctDNA, cfDNA, cell-free tumor DNA, liquid biopsy, colorectal cancer, minimal residual disease, DNA mutation, next-generation sequencing, UMI-mediated error suppression, clonal hematopoiesis

## Abstract

Circulating tumor DNA (ctDNA) is a promising biomarker, reflecting the presence of tumor cells. Sequencing-based detection of ctDNA at low tumor fractions is challenging due to the crude error rate of sequencing. To mitigate this challenge, we developed ultra-deep mutation-integrated sequencing (UMIseq), a fixed-panel deep targeted sequencing approach, which is universally applicable to all colorectal cancer (CRC) patients. UMIseq features UMI-mediated error correction, the exclusion of mutations related to clonal hematopoiesis, a panel of normal samples for error modeling, and signal integration from single-nucleotide variations, insertions, deletions, and phased mutations. UMIseq was trained and independently validated on pre-operative (pre-OP) plasma from CRC patients (*n* = 364) and healthy individuals (*n* = 61). UMIseq displayed an area under the curve surpassing 0.95 for allele frequencies (AFs) down to 0.05%. In the training cohort, the pre-OP detection rate reached 80% at 95% specificity, while it was 70% in the validation cohort. UMIseq enabled the detection of AFs down to 0.004%. To assess the potential for detection of residual disease, 26 post-operative plasma samples from stage III CRC patients were analyzed. From this we found that the detection of ctDNA was associated with recurrence. In conclusion, UMIseq demonstrated robust performance with high sensitivity and specificity, enabling the detection of ctDNA at low allele frequencies.

## 1. Introduction

Circulating tumor DNA (ctDNA) has emerged as a powerful biomarker reflecting the presence of tumor cells, tumor burden, and patient outcome across multiple cancers, including colorectal cancer (CRC) [1,2]. Mutation-based tumor-informed sequencing methods are among the most widely used ctDNA detection approaches [3,4,5,6,7]. These methods can be categorized into two types: (1) the bespoke approach, where a unique assay targeting patient-specific mutations is designed for each patient [5,8,9], and (2) the fixed approach, where the same assay, usually a cancer-specific capture panel, is applied to all patients [7,10,11]. The strength of a bespoke approach is a high ratio of tumor DNA markers to the number of analyzed genomic base pairs. However, this comes at the expense of having to design a new assay for each patient and the prolonged turnaround time associated with this. Although a fixed approach often has a lower ratio of tumor DNA markers to the number of analyzed genomic base pairs, compared to a bespoke approach, it has the advantage of convenience and shorter turnaround times.

For variant calling in circulating cell-free DNA (cfDNA), most studies have focused solely on single-nucleotide variants (SNVs) [11,12,13], a combination of SNVs and insertions and deletions (INDELs) [7], or, more recently, phased variants [14]. Due to the higher complexity of phased variants, and possibly also INDELs and multi-nucleotide variants (MNVs), these variants often have a low error rate, making them attractive targets for the detection of low-frequency variants [14]. Furthermore, incorporating several independent mutational classes into a ctDNA caller may enhance the tumorigenic signal and increase the sensitivity of ctDNA detection approaches.

Here, we present a tumor-informed fixed-panel strategy, a termed targeted **u**ltra-deep **m**utation-**i**ntegrated **seq**uencing strategy (UMIseq). UMIseq leverages unique molecular identifiers (UMIs) for comprehensive error correction and employs a panel of normal (PON) for mutation-specific error modeling. To exploit the information from all the somatic tumor-specific mutations identified within the regions covered by the fixed panel, UMIseq integrates the signal from all mutations into a single measure. Bayesian inference is used to assess if this signal is higher than expected based on the PON, i.e., if the sample is ctDNA-positive. UMIseq leverages information from various classes of mutations: SNVs, INDELs, MNVs, and phased variants. We conduct a rigorous assessment of the performance of the UMIseq approach using a comprehensive training and validation cohort design, including an evaluation of minimal residual disease (MRD) detection in post-operative (post-OP) plasma samples from stage III CRC patients.

## 2. Results

### 2.1. The UMIseq Method and Study Design

We first evaluated UMIseq (method overview in Figure 1a) by characterizing the general error profile across all positions in the capture panel and then by assessing the analytical performance using a set of synthetic mixture samples with a varying concentration of tumor DNA mixed in normal DNA (Figure 1b). Next, we trained UMIseq on a cohort of plasma from healthy individuals (*n* = 37) and pre-OP plasma samples from patients with stage I–IV CRC (*n* = 126). The performance of UMIseq was validated in an independent cohort of pre-OP plasma samples from patients with stage I–III CRC (*n* = 209), minimally invasive pT1pN0 cancers (*n* = 29), and non-invasive colorectal adenomas (*n* = 17), as well as plasma from healthy individuals (*n* = 24) (Figure 1b). Finally, the utility of UMIseq to predict minimal residual disease was assessed in post-OP samples from stage III CRC patients (*n* = 26) (Figure 1b).

UMIseq of plasma samples resulted in median UMI consensus sequencing depths of 8367 (IQR 9225) and 10,353 (IQR 9191) for the training and the combined validation and MRD cohorts, respectively (Appendix A). Of note, the median sequencing depths are partly determined by the cfDNA input. The median plasma cfDNA conversion efficiency was 88% (IQR 47%) and 83% (IQR 52%) for the training and the combined validation and MRD cohorts, respectively (Appendix A). Comparable median depths were obtained from UMIseq on PBMC samples used for filtering mutations related to clonal hematopoiesis of indeterminate potential (CHIP) (Appendix A). The clinicopathological characteristics of the training and validation can be seen in Table 1. The applied capture panel was designed to capture at least one mutation in all patients. In practice, 93% (353/381) of patients had at least two mutations within the capture panel (median three mutations per patient, IQR 2) (Table 1, Appendix A).

### 2.2. Error Characterization of UMIseq

To form a PON, UMIseq was applied to cfDNA from healthy individuals (*n* = 46) and sequenced to a median depth of 9086 (IQR 5464) (Appendix A). As the PON samples are not expected to carry any cancer-related mutations, they facilitated an exploration of the error profile associated with UMIseq. All non-reference base calls at each genomic position targeted by the capture panel were considered to be errors. The absolute number of errors in each plasma sample was positively correlated with the sequencing depth (r = 0.79; *p* < 0.0001, Pearson’s correlation).

The most frequently occurring errors were G > T transversions and C > T and G > A transitions (Figure 2a). More than 90% of the C and G sites within the panel had C > T, G > T, and G > A substitutions with an error rate above 0.001%, and for approximately 25% of the sites, the noise was higher than 0.01% (Figure 2b). In contrast, between 75% and 80% of T > G, A > C, C > G and G > C substitutions had an error rate of less than 0.001% (Figure 2b). Deletions, and in particular insertions, displayed low error rates (Figure 2b).

Further evaluation including the trinucleotide context of each substitution revealed that the error rate of certain substitutions was highly affected by the neighboring 3′ or 5′ nucleotides, while others were not. Regardless of the trinucleotide context, T > G and C > G substitutions generally had a low error rate, whereas C > T substitutions had the highest error rates. The error rates of C > T substitutions were particularly high when the 3′ base was G, i.e., N (C > T) G, independent of the 5′ base (Figure 2c).

### 2.3. Assessment of the Theoretical Limit of Detection

Based on the error rates observed when applying UMIseq to cfDNA from healthy individuals, we predicted the limit of detection (LOD) for SNVs and single-base INDELs. This revealed that INDELs had a significantly different, and generally lower, LOD distribution compared to that of SNVs (*p* < 0.0001, Kolmogorov–Smirnoff test) (Figure 2d). Notably, 48% of CRC patients exhibited at least one deletion and 22% had at least one insertion within the capture panel (Figure 2e). In addition, MNVs were observed in 9% of CRC patients. To assess the sensitivity of UMIseq based solely on SNVs or a combination of SNVs, INDELs, and MNVs, we generated 25 in silico patient mutational catalogs by the random selection of SNVs and non-SNVs (INDELs and MNVs) from the collective mutational catalog of all patients in the study (as described in Appendix B).

Next, we generated synthetic samples for each in silico patient by Poisson sampling 20,000 counts using a Poisson rate λ = cAF for each mutation. This process was carried out at four distinct cAFs: 0.1667%, 0.0556%, 0.0185%, and 0.0062%. For each patient, two integrated *s*-scores were generated: one including both SNV and non-SNV mutations and one including only the SNV mutations (i.e., INDELs and MNVs were removed from the mutation catalog). Inclusion of the non-SNVs increased the likelihood of correctly assigning a ctDNA-positive label to samples across all cAFs, with the tendency increasing with a higher cAF (Figure 2f).

**Figure 2 ijms-25-04252-f002:**
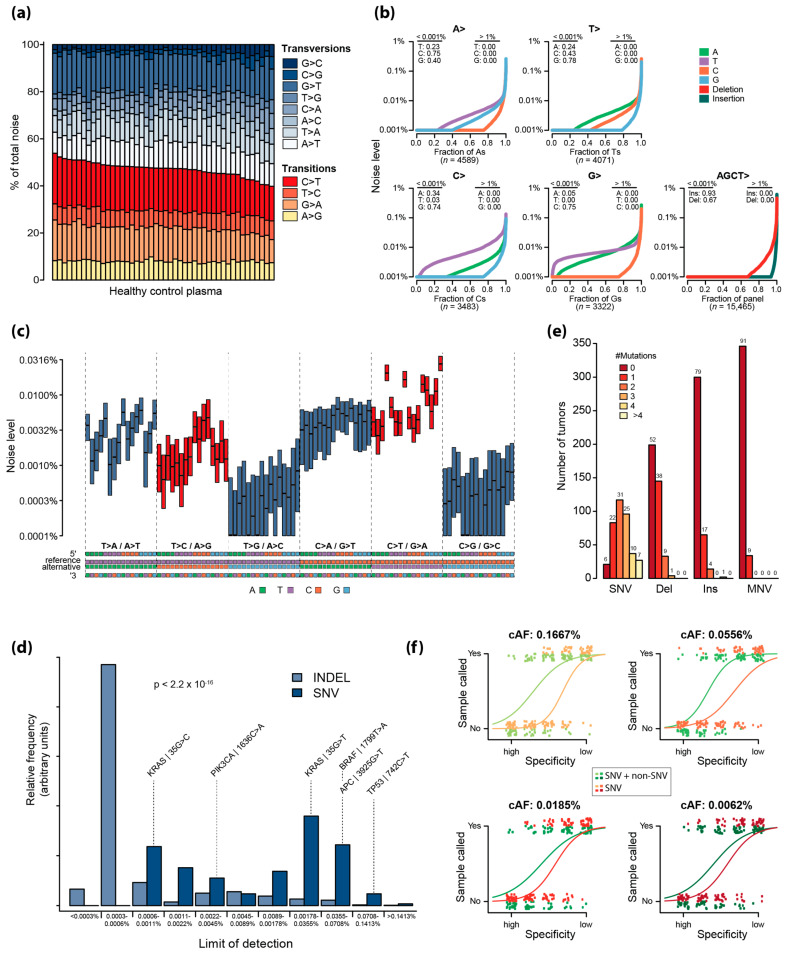
(**a**) Noise distribution shown as relative frequency of single-nucleotide errors observed in healthy control plasma (*n* = 46) across the target regions (15,465 bp). (**b**) The mean noise across healthy plasma controls for each single-base substitution, insertion, and deletion. The fraction of sites with a noise level below 0.001% and above 1% are indicated. (**c**) Boxplot of the noise level (median and IQR) of single-base substitutions according to their trinucleotide context. (**d**) Predicted LODs of SNVs (dark blue) and INDELS (light blue) at 99% specificity. The LODs of common CRC mutations are annotated. (**e**) The frequency of SNVs, deletions, insertions, and multi-nucleotide variants (MNVs) in tumors from all patients (*n* = 381). The fraction (in percentage) is indicated at the top of each bar. (**f**) Simulated probabilities of calling a sample positive with or without non-SNV alterations at various circulating allele frequencies and specificities.

### 2.4. Analytical Sensitivity of UMIseq

To evaluate the analytical sensitivity of UMIseq, we mixed tumor DNA from 36 patients to create a compound sample (Sample A) with 95 ground-truth mutations within the target regions of the capture panel (the foreground mutations) (Figure 3a). Sample A was serially diluted with normal PBMC DNA (Sample X) to create five synthetic ctDNA samples (Samples B–F) with decreasing tumor DNA fractions.

Comparison of the error rate of the foreground mutations (*n* = 95) to that of recurrent CRC mutations (*n* = 275) occurring in the COSMIC database [15] showed a high similarity (*p* = 0.2928, Kolmogorov–Smirnov test), indicating that the foreground mutations were representative of CRC mutations (Figure 3b). Of note, the error rate across the entire panel (background) was lower than that of both the foreground and the COSMIC mutations (*p* < 0.0001, Kolmogorov–Smirnoff test), which may hint at an increased underlying error rate of recurrent mutations in CRC.

There was a good correlation between the expected AFs of the foreground mutations in Sample A and the observed AFs from UMIseq (r = 0.9878, *p* < 0.0001, Pearson’s correlation) across an AF span from 0.1% to almost 100% (Figure 3c). The strong correlation between expected and observed AFs was independent of whether the mutations occurred in COSMIC (*n* = 36; r = 0.9934, *p* < 0.0001) or not (*n* = 59; r = 0.9011, *p* < 0.0001). By grouping the mutations observed in samples A through F based on their observed AF (Figure 3a), we assessed the performance of UMIseq as a function of AF. Remarkably, UMIseq displayed an area under the curve (AUC) above 0.95 for the precision–recall and ROC estimates down to AFs of 0.05% (Figure 3d). At an AF below 0.05%, the absolute number of reads supporting the mutations approached a single read, i.e., the mean number of reads supporting each mutation was 1.2 reads in the AF range of 0–0.01%. This indicates that at very low AFs, sampling effects, rather than the error rate of UMIseq, determine whether ctDNA is detected.

### 2.5. Training of UMIseq for ctDNA Detection Using Pre-OP Plasma

The UMIseq algorithm was trained to call ctDNA in plasma samples with 95% specificity using a cohort of pre-OP plasma samples from patients with stage I–IV CRC (*n* = 126; stage I: 32; stage II: 59; stage III: 28; stage IV: 7) and plasma from a control group of healthy individuals (*n* = 37). ROC curve analysis showed an AUC of 0.90, demonstrating that UMIseq has both high sensitivity and specificity (Figure 4a) and that integrating signal from all available mutations in UMIseq increases the performance compared to a strategy where only a single mutation is used (ROC AUC 0.83, Appendix A).

We estimated the LOD of UMIseq as the minimum cAF required to call a sample ctDNA-positive with 95% specificity. This corresponded to a UMIseq score cutoff α = 0.9797, which was used henceforth.

An inverse correlation between the LOD and sequencing depth was seen, with the median LOD reaching 0.011% (IQR 0.029%) and 10% of estimated LODs were below 0.001% at a 20,000× depth (Figure 4b). Overall, the sensitivity of UMIseq was 80% (101/126) among the pre-OP samples.

The detection rate was 69% (22/32) in stage I, being even higher in stage II (85%; 50/59) and stage III (79%; 22/28), and it was highest in stage IV (100%; 7/7) (Figure 4c). The ctDNA detection rate increased with the pathological T (pT) category (pT2: 68%, pT3: 83%, pT4: 100%), while there was no correlation to the pathological N (pN) category (pN0: 79%, pN1-2: 79%) (Figure 4d).

### 2.6. Analytical Robustness of ctDNA Detection Using UMIseq

To evaluate the analytical robustness of UMIseq, we assessed ctDNA detection in three settings using paired samples. Firstly, we tested the performance of UMIseq against an orthologous method—single mutation digital droplet PCR (ddPCR)—using two plasma aliquots (*n* = 11) from the sample blood collection. We found that 91% (10/11) of samples analyzed with UMIseq were ctDNA-positive, while 64% (7/11) were ctDNA-positive by ddPCR. The estimated cAFs of samples classified as ctDNA-positive by both methods demonstrated a near-perfect correlation (r = 0.9950, Pearson’s correlation) (Figure 4e).

Secondly, we tested the reproducibility of the entire workflow (from cfDNA purification through UMIseq calling) by procuring two plasma aliquots from 19 randomly chosen patients and repeating the entire workflow from cfDNA extraction to UMIseq calling. For 84% (16/19) of the sample pairs, the UMIseq ctDNA classifications agreed. Additionally, a strong correlation was observed between the estimated cAFs in pairs where both samples were ctDNA-positive (r = 0.9952, Pearson’s correlation) (Figure 4f).

Lastly, we tested the repeatability of UMIseq by splitting 49 randomly chosen cfDNA samples in two aliquots with identical amounts of cfDNA which were then subjected to UMIseq analysis. Similarly to the reproducibility analysis, we observed consistent ctDNA classification in 84% of them (41/49) and a strong correlation between the cAFs of the split samples (r = 0.9882, Pearson’s correlation) (Figure 4g).

### 2.7. Validation of UMIseq

Having determined the UMIseq score threshold yielding 95% specificity for ctDNA detection in the training cohort, we next sought to validate whether this threshold would yield a similar specificity in an independent set of plasma from healthy individuals (*n* = 24). At the predefined UMIseq score threshold, the specificity estimates were indistinguishable from those of the training cohort (Appendix A and Appendix B).

We next conducted an independent validation of the UMIseq sensitivity on pre-OP plasma from stage I–III CRC patients (*n* = 209; stage I: 52, stage II: 108, stage III: 49). Overall, the sensitivity of UMIseq was 70% (146/209) in the validation cohort. Stratifying the ctDNA detection rate based on UICC (Figure 5a) and the pT and pN categories (Figure 5b) confirmed the association between the pT category and ctDNA detection (pT2: 56%; pT3: 75%; pT4: 81%) previously observed in the training cohort.

To evaluate the potential for ctDNA detection in very low-stage cancers and premalignant disease, we applied UMIseq to pre-OP plasma obtained from patients diagnosed from fecal immunochemical test screening with minimally invasive CRC (pT1pN0) and colorectal adenomas. UMIseq detected ctDNA in the plasma of 31% (9/29) of the patients with pT1pN0 tumors but in none of the patients with adenomas (Figure 5c).

### 2.8. Association between pT, Tumor Size, and ctDNA

Pre-OP cfDNA samples from the cancer patients from the training and the validation cohorts (*n* = 364) were used to investigate associations between analytical and clinical factors and ctDNA. We found no association between the plasma sequencing depth or conversion efficiency and ctDNA detection. A trending association was observed between the number of tumor-specific mutations within the capture panel and ctDNA detection (*p* = 0.0967, OR 1.14).

Next, we investigated the ctDNA levels in relation to the depth of invasion, spread to regional lymph nodes, and tumor size. The estimated UMIseq cAFs of ctDNA-positive samples ranged from 0.004% to 71.9% (median 0.095%, IQR 0.271%). In general, the median cAF increased with the pT category, except for pT4, which only displayed an increased cAF compared to that of pT1 (median cAF: pT1 = 0.023% [IQR 0.013%]; pT2 = 0.070% [IQR 0.087%]; pT3 = 0.155% [IQR 0.331%]; pT4 = 0.070% [IQR 0.305%]) (Figure 5d). There was no significant difference between the median cAF and the pN category (median cAF: pN0 = 0.105% [IQR 0.296%]; pN1 = 0.064% [IQR 0.137%; pN2 = 0.105% [IQR 0.235%]) (Figure 5e). Furthermore, in a multivariable logistic model, pT but not pN was strongly associated with ctDNA detection with the odds ratio of detecting ctDNA being 3.41 in pT2 compared to in pT1 (*p* < 0.001) and as much as 20.24 in pT4 (*p* < 0.001) (Appendix A), an observation that complemented the pT-stratified and pN-stratified ctDNA detection rates (Figure 4c and Figure 5b). Even when adjusting for tumor size (longest diameter), a similar, but less strong, trend was observed (Appendix A).

In general, cAFs increased with larger tumor size (Figure 5f). Furthermore, tumor size was strongly associated with the pT category, with more than 95% of pT1 tumors being classified as “very small tumors” while pT3 and pT4 tumors predominantly were “large tumors” and “very large tumors” (Figure 5f) (*p* < 0.0001, Fisher’s exact test). Lastly, the tumor size of ctDNA-negative patients (median 23.0 mm) was significantly smaller than the tumor size of ctDNA-positive patients (median 40.0 mm) (*p* < 0.0001, Wilcoxon rank sum test) (Figure 5g).

### 2.9. Assessment of Clonal Hematopoiesis of Indeterminate Potential

As part of the UMIseq pipeline, potential mutations related to CHIP are identified in patient-matched normal PBMC DNA and excluded from the mutational catalogue and hence do not contribute to the UMIseq score of that patient. Since the PBMC DNA in this study was sequenced to the same depths as the plasma cfDNA (Appendix A), it allowed us to comprehensively evaluate the frequency of CHIP within the capture panel, i.e., in the most recurrently mutated regions in CRC. We found that 3.4% (13/381) of patients had CHIP mutations, comprising 14 distinct mutations in the genes *APC*, *TP53*, *TCF7L2*, *KRAS*, and *SOX9* (Figure 5h). In general, the cAF observed in PMBC DNA was higher than the cAFs observed in the cfDNA, with only four out of the fourteen mutations having a higher cAF in cfDNA. While the cAFs in PBMCs were generally low (less than 1%), *APC* was the only gene with cAFs less than 0.1%.

### 2.10. UMIseq Applied for Minimal Residual Disease Detection

We investigated the ability of UMIseq to detect MRD in post-OP plasma samples collected after operation and before the initiation of adjuvant chemotherapy from patients with stage III CRC (*n* = 26). The recurrence rate in this cohort was 31% (9/26). UMIseq detected ctDNA in 33% (3/9) of recurrence patients (Figure 5i) and post-OP ctDNA detection were associated with an increased risk of recurrence (*p* = 0.08, log-rank test) (Figure 5j). ctDNA was detected in 12% (2/17) of non-recurrence patients, which translates to an apparent post-OP specificity of 89% (Figure 5i). This corresponds to a positive predictive value of 60% and a negative predictive value of 71%.

## 3. Discussion

The identification of cancer through ctDNA analysis has several clinical implications, and, currently, the use of ctDNA analysis to detect and monitor residual disease after the operation is extensively investigated [3,5,16]. For successful clinical implementation in this setting, it is crucial for ctDNA analysis to exhibit analytical robustness and allow for the detection of low tumor burden (high sensitivity), while simultaneously minimizing the occurrence of false positives (high specificity). Additionally, the cost and simplicity of the analysis are inevitable factors that impact the utilization of ctDNA analysis. In this study, we introduced UMIseq. By using a fixed panel, designed to be applicable to all CRC patients, UMIseq provides a simple and time-efficient approach, resulting in a single, sample-level call.

The key to sensitive ctDNA detection at a high specificity is a high signal-to-noise ratio. This can be achieved by increasing the signal, lowering the noise, or both. On one end, methods that aim at lowering the error rate, such as Duplex sequencing, achieve an error rate approaching 10^−7^. However, Duplex sequencing suffers from the inefficient recovery of both original strands, which occurs in a minority (usually 20–25%) [10,17] of DNA fragments, thus limiting the sensitivity. On the other end, WGS approaches aim to increase the signal by integrating the genome-wide signal for ctDNA detection. Performance comparisons between the different approaches are challenging due to the lack of adequate benchmarking studies, leaving comparisons to be made between studies and at the cohort level. This has multiple weaknesses, e.g., that plasma volumes, sample processing, and cohort compositions are different.

The analytical sensitivities observed in plasma samples are, a bit surprisingly, quite similar for targeted and WGS approaches, with the lowest detected AFs often being in the range 10^−4^ to 10^−5^, indicating that in practice both targeted sequencing and WGS strategies have their own limitations [11,12,14,18,19]. Presumably, targeted strategies that employ comprehensive error correction through single-strand UMI or Duplex sequencing are limited by the amount of signal captured for ctDNA detection, while WGS approaches are limited by their error rate despite comprehensive error modeling and correction. UMIseq enabled ctDNA detection down to a cAF below 10^−4^, with the lowest observed cAF in a ctDNA-positive sample being 0.004% and a theoretical LOD below 0.001% in 10% of the estimated LODs at a 20,000× sequencing depth.

The pre-OP sample level sensitivity of UMIseq is comparable to that of widespread commercial tests such as the Signatera test, which are reported as 40%, 92%, and 90% for CRC UICC stage I, II, and III, respectively [20]. Hence, UMIseq outperforms Signatera for the lower stage, but not for higher stages, although differences in cohort compositions hinder a definite comparison.

Similarly to findings in other studies [13,19,21], the error rate of UMIseq was highly dependent on nucleotide substitution and trinucleotide context. Particularly, C > T substitutions in the context of N(C > T)G were error-prone, while T > G and C > G substitutions, regardless of trinucleotide context, had a low error rate. In general, INDELs, especially insertions, had a lower error rate than that of SNVs, which was also reflected in the theoretical LOD calculation. Although these observations are based on a fixed hybrid-based capture panel, similar trends have been observed from ddPCR [21] and are expected to be generalizable to other approaches as well. This suggests that a preference towards substitutions with a low trinucleotide-context error rate, and the inclusion of non-SNV targets could further enhance the sensitivity of ctDNA detection.

A potential challenge to maintaining the specificity of ctDNA testing is the risk of false positive ctDNA calls by misinterpreting CHIP as ctDNA mutations [22,23,24]. Studies employing the deep sequencing of PBMCs from healthy individuals suggest that low-frequency CHIP mutations (<0.1%) are present in up to 92% of patients [22,24]. When utilizing a tumor-informed strategy for ctDNA detection, only CHIP-related mutations that overlap with tumor-specific variants can contribute to false positive ctDNA calls. From the tumor-informed UMIseq analysis of PMBC DNA, we identified and excluded CHIP-related mutations in 3.4% (13/381) of the patients. These results demonstrate that while CHIP-related mutations can still be an issue for a tumor-informed strategy, it is far less likely to result in a false positive call than when using a tumor-agnostic approach. To minimize the additional cost and labor associated with CHIP analysis, it may be feasible to restrict CHIP analysis to patients with a ctDNA-positive sample.

Previous studies have demonstrated that the ctDNA detection rate increases with the UICC stage and tumor size [25,26]. While our study supports these observations, we find that this correlation is primarily driven by the pathological T stage. This is corroborated by the lower detection rate of pT1pN0 tumors and the absence of ctDNA in patients with non-invasive colorectal adenomas. In both the training and validation of UMIseq, we surprisingly found a complete lack of association between ctDNA detection and lymph node tumor infiltration (pN stage), although we cannot rule out that a small association would be identified in a larger patient population. Similar observations have previously been noted in smaller cohorts of gastric, lung, and colorectal cancers [27,28]. One interpretation of this is that the additional tumor burden from regional lymph node involvement does not contribute to increased blood ctDNA levels, suggesting that tumor cells residing in the lymph nodes shed very limited amounts of cfDNA into the bloodstream. Possibly, this can be attributed to an abundance of immune cells in the lymph nodes, which degrade cell residues, including any ctDNA released by the tumor cells. A potential consequence could be a reduced ability to detect residual tumor cells residing in lymph nodes after operations, which is consistent with recent observation in a study of colorectal metastases [29].

We developed UMIseq with the aim of providing a sensitive and uniform approach for the detection and surveillance of residual disease in CRC patients after an operation. In addition to thoroughly evaluating the performance of UMIseq, we investigated the potential for MRD detection in post-OP plasma samples. This revealed a sensitivity of 33%, a specificity of 89% for post-OP ctDNA detection. We note that both ctDNA-positive non-recurrence patients received adjuvant chemotherapy, which may have eliminated residual cancer cells left after the operation. While UMIseq did already enable the detection of MRD immediately after the operation, the ctDNA detection at this timepoint is possibly challenged by a very low tumor burden and a surge in release of trauma-related cfDNA [30]. Through serial surveillance, ideally using plasma samples collected after the end of adjuvant chemotherapy, the sensitivity is expected to improve. The lower specificity, compared to the specificity observed in the training and validation cohort, is expected in stage III CRC as this patient group usually receives adjuvant chemotherapy after the collection of the post-OP sample [3,20,31], which was also the case for the two ctDNA-positive non-recurrence patients. Here, we explored MRD detection in a small cohort of stage III CRC patients, and further studies are required to assess the clinical utility of UMIseq in the post-operative setting across all stages.

There are limitations to this study. First, the targeted panel applied for UMIseq was specifically designed for CRC patients and can therefore not be applied to other cancer types in its current design. However, by designing a new capture panel, UMIseq could be easily adapted for other cancer types while keeping the protocol for sample preparation and the bioinformatic pipeline consistent. The required size of such a panel should be altered according to cancer type and may result in increased sequencing cost.

The panel designed for this study was tailored for the identification of small somatic mutations. As off-target persists as a relatively large fraction (~15%) within the sequencing data, there is potential to leverage this information and gain insight into patient-specific copy number alterations. This could provide a more comprehensive patient-specific tumor catalogue for improved ctDNA detection. While we demonstrate the overall robustness of UMIseq, the reproducibility and repeatability assessment were conducted within a single laboratory, preventing evaluations of the inter-laboratory robustness of UMIseq.

In summary, we have presented UMIseq as a sensitive and universal approach for ctDNA detection, where the same laboratory workflow and capture panel are applied to all CRC patients. In this study, we thoroughly assessed the performance of UMIseq using pre-operative plasma samples and investigated the potential of UMIseq for MRD detection. The potential application of this approach is aimed at detecting MRD, treatment response, and monitoring for recurrence, all of which are not limited to CRC but could also be applied to other cancers, provided an appropriate capture panel for the given cancer type can be designed.

## 4. Materials and Methods

### 4.1. Study Population

For this study, 383 CRC patients and 107 healthy controls were recruited and divided into a training cohort, a validation cohort, and a panel of normal (PON) control cohort. Two patients (0.52%) were excluded because none of the mutations in the tumor overlapped with the UMIseq capture panel. The training cohort included 37 healthy individuals and 126 stage I–III patients (stage I (pT2pN0cMO): *n* = 32; stage II (pT3-4pN0cM0): *n* = 59; stage III (pT1-4pN1-2cM0): *n* = 28; stage IV: *n* = 7). The validation cohort consisted of 24 healthy individuals and 209 stage I–III patients (stage I (pT2pN0cMO): *n* = 52; stage II (pT3-4pN0cM0): *n* = 108; stage III (pT1-4pN1-2cM0): *n* = 49), 29 patients with polyp cancers (pT1pN0cM0), and 17 patients with colorectal adenomas. Post-OP plasma samples (*n* = 26) from stage III patients with at least 30 months of radiological follow-up, or radiological-confirmed local or distant recurrence were used for assessing the performance of UMIseq to detect MRD after operation. The PON consisted of plasma samples from 46 healthy individuals.

From all patients, tumor tissue, peripheral blood mononuclear cells (PBMCs), and an 8 mL pre-operative (pre-OP) plasma sample were collected. Healthy individuals were anonymously recruited through the blood bank at Aarhus University Hospital or through the Danish colorectal cancer screening program. From healthy individuals, 8 mL of plasma was collected.

### 4.2. Sample Collection and Processing

Details on tumor and blood sample collection, DNA extraction, library preparation, capture of target regions, and sequencing are described in Appendix B. In brief, tumor DNA and normal DNA from PBMCs were used to establish mutational profiles for each patient, as well as identify mutations associated with clonal hematopoiesis. For 12 patients diagnosed with synchronous tumors, we used the mutational profiles from both tumors in the ctDNA analysis. cfDNA from 8 mL plasma and normal DNA from each patient were analyzed by targeted sequencing using a custom panel based on hybridization capture, which included the most frequently mutated genomic regions observed in patients with CRC [15] (15,465 bp, Appendix A). All samples (tumor, normal, and cfDNA) were sequenced by paired-end (2 × 150 bp) sequencing using the Illumina NovaSeq platform. Details can be found in Appendix B.

### 4.3. Plasma Variant Calling and Estimation of Circulating Allele Frequency

Variants were called in cfDNA and PBMC DNA using the Shearwater algorithm [32]. Shearwater models the counts on the forward and backward strands with a beta-binomial model to compute a Bayes factor for each possible SNV and uses multiple samples to estimate local error rates and dispersion. Here, the implementation was abstracted to also include INDELs, complex MNVs, as well as phased mutations (Appendix A). To this end, every mutation, regardless of its class, was represented as a read count vector of length four, consisting of forward- and backward-strand alternative counts (ALT and alt), and forward and backward non-alternative counts (^ALT and ^alt). For mutations spanning more than one nucleotide (most INDELs and MNVs), the mean counts across the locus were used.

The ALT/alt and ^ALT/^alt counts were extracted from the UMI consensus BAM files. For both PBMC and cfDNA samples, the analyses were tumor-informed, i.e., counts were only extracted for mutations reported in the tumor VCF file of the matching patient.

The bbb function from the R package deepSNV [33] was used to calculate a mutation score, *m*, for each mutation, which equals the Bayes factor generated by bbb using the parameters as follows: model = ‘AND’ and prior = 0.5. The plasma samples of the PON cohort were processed in parallel with all other plasma samples and used to estimate the dispersion rho. The product of the Bayes factors of each sample’s mutation catalog was calculated to generate an integrated score, *s*, representing the likelihood of the sample given being negative over the likelihood of the sample given being positive.

To mitigate the possibility that the noise structure of a particular sample deviated significantly from that of the PON, the integrated score was transformed into the UMIseq score, S. The UMIseq score represents a one-sided empirical p-value expressing how extreme the observed score *s* is compared to all other possible *s*-scores that could be generated in the sample using n mutations. To do this, a distribution of sample scores, **S^n^,** was generated from K random scores, i.e., sample scores from random mutation catalogs of length n sampled from all possible positions of the panel. The UMIseq score, S, was then calculated as the normalized rank of the observed score, *s*, in **S^n^**, specifically S = 1 − (r + 1)/(K + 1), where K = 100,000, and r is the number of random scores larger than *s*. Plasma samples with a score above a fixed threshold, α, resulted in the sample being called ctDNA-positive. As an estimate of the mutational burden observed in a plasma sample, the mutational circulation allele frequency (cAF) was calculated as the sum of reads supporting the set of patient-specific tumor-informed mutations, divided by the sum of all reads across the mutated loci.

### 4.4. Blacklisting of SNVs and INDELs

A total of 69 mutations were blacklisted as they were recurrent in plasma cfDNA from healthy individuals. Details can be found in Appendix B.

### 4.5. Flagging of Mutations Associated with Clonal Hematopoiesis

UMIseq analysis of PBMC samples matched to each patient was performed to explore if any of the mutations identified in the tumor also co-occurred in hematopoietic cells (clonal hematopoiesis of indeterminate potential: CHIP). Details on this can be found in Appendix B.

### 4.6. Limit of Detection Calculation

The limit of detection (LOD) of individual SNVs and INDELs within the capture panel was assessed to explore their differences in LOD. A distribution of sample UMIseq LODs was similarly estimated. Details can be found in Appendix B.

### 4.7. In Silico Estimation of the ctDNA Detection Probabilities

Details on in silico estimation of ctDNA detection probabilities at various cAFs can be found in Appendix B.

### 4.8. Recurrent COSMIC Mutations

Details on identification of recurrent COSMIC mutations within the UMIseq capture panel can be found in Appendix B.

### 4.9. Analytical Sensitivity Analysis Using a Synthetic Mixture

Details on assessment of the analytical sensitivity can be found in Appendix B.

### 4.10. Tumor-Informed Model Training of the UMIseq Algorithm

To generate a UMIseq score threshold, α, for calling a plasma sample ctDNA-positive, UMISeq was applied to a training cohort consisting of pre-OP plasma from CRC patients (*n* = 126) and healthy controls (*n* = 37). To utilize the full diversity of the healthy control samples and the training cohort mutation catalog (*n* = 276 unique mutations), 25 Monte Carlo simulations were made as follows: UMIseq scores, S, from 100 randomly selected CRC cfDNA samples were calculated using their matching tumor catalog and used as positive labels. One hundred UMIseq scores representing non-cancer samples (negative labels) were calculated by applying in silico-generated mutation catalogs to randomly selected healthy control cfDNA samples. Each in silico catalog was generated by sampling n mutations without replacement from the 276 mutations in the training cohort with probabilities equal to their relative frequency. The number of mutations (n) to sample in each catalog followed the distribution of mutations per patient in the training cohort.

For each of the 25 simulations, the receiver operator characteristic (ROC) statistics, as well as the UMIseq S score resulting in a 5% FPR (corresponding to 95% specificity), were calculated. The mean of the UMIseq scores yielding 5% FPR across the 25 simulations was then used as a fixed cutoff, α, for calling plasma samples ctDNA-positive in the validation cohort, in the orthologous method, in the reproducibility, and in repeatability tests, as well as for estimating the LOD of UMIseq.

### 4.11. Assessment of UMIseq Robustness

For a subset of samples, the robustness of UMIseq was assessed by (i) comparison to an orthogonal method (ddPCR), (ii) by procuring, processing, and analyzing a new plasma aliquot, and iii) by generating two cfDNA libraries from the same cfDNA sample. Details can be found in Appendix B.

### 4.12. Validation of Specificity

cfDNA from 24 plasma samples obtained from healthy individuals were extracted and subjected to UMIseq. These were used as independent controls to validate the 95% specificity that was set in the model training. We repeated the procedure for generating negative sample sets as described for the training cohort (see above). Random mutation sets (*n* = 100) were sampled from the training cohort tumor catalog according to their individual frequencies and used to generate 100 UMIseq scores, S, in the 24 validation controls. The specificity was calculated as the fraction of scores resulting in a negative call. This was repeated 25 times, and a *t*-test was applied to test the difference in mean specificity between validation and training.

### 4.13. Statistical Considerations and Calculations

Statistical tests were used as appropriate and as described in the text. All calculations were performed using R (3.5.1) in the Rstudio environment (1.1.456) and with packages deepSNV [33] (1.28.0), genomicranges (1.34.0), rsamtools (1.34.0), abind (1.4.5), dplyr (1.4.3), survival (3.1), and reshape2 (1.4.4) installed under the conda (23.3.1) environment.

## Figures and Tables

**Figure 1 ijms-25-04252-f001:**
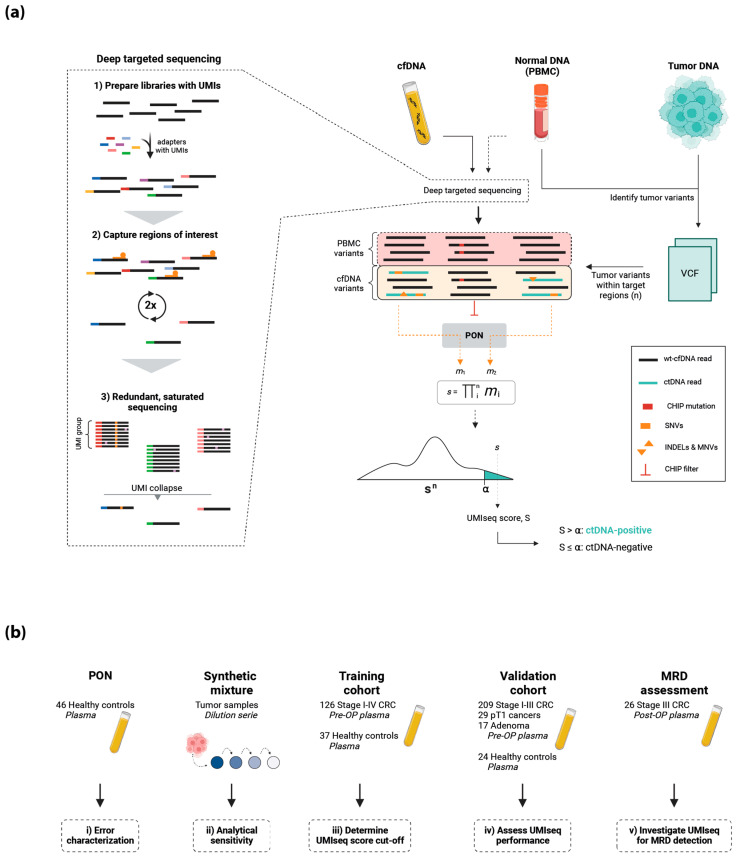
(**a**) The workflow for deep target sequencing (left panel) includes library preparation using UMI-containing sequencing adapters; two consecutive rounds of hybridization-based capture of genomic regions frequently mutated in CRC; and redundant, saturated sequencing to ensure the creation of UMI families with at least 3 reads. Each of these are finally collapsed into a consensus read, thereby eliminating random sequencing errors. cfDNA from plasma, and optionally also DNA from PBMCs, are subjected to deep targeted sequencing (right panel). DNA extracted from the tumor and PBMCs are used to identify tumor-specific variants (n), which are then searched for in the consensus reads from deep targeted sequencing of cfDNA and PBMC DNA. Variants present in the PBMC DNA are classified as CHIP mutations and excluded. For each of the n variants observed in cfDNA, a mutation score, *m*, is calculated based on the noise and dispersion observed in the panel of normal (PON). By integrating the signal from all *m* scores, a sample score, *s*, is calculated. To mitigate the possibility that the noise structure of a particular sample may deviate significantly from that of the PON, the integrated sample score, *s*, is transformed into the UMIseq score, S. To do this, a sample-specific score distribution, **S^n^**, is generated by calculating UMIseq scores in the cfDNA sample from 100,000 in silico random mutation catalogs harboring the same number of mutations as in the sample. S is then calculated as the rank of the integrated score of the tumor mutation catalog in the distribution, **S^n^**. Samples with a UMIseq score above a fixed threshold, α, are deemed to be ctDNA-positive. (**b**) Overview of the cohorts used in this study. Figure created with Biorender.com (accessed on 25 January 2024).

**Figure 3 ijms-25-04252-f003:**
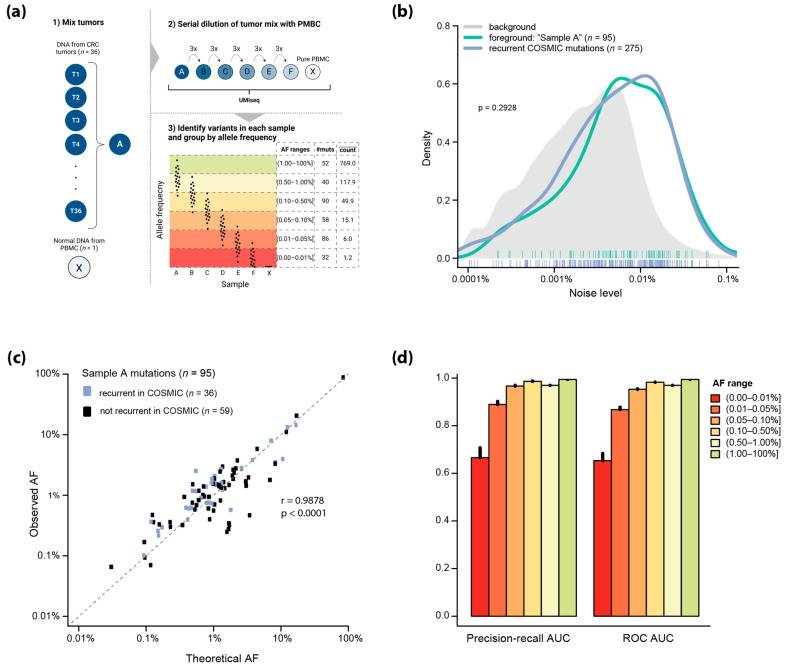
(**a**) Schematic overview of the strategy for creating and analyzing artificial sample sets with known variants (foreground: *n* = 95 mutations identified in 36 tumor samples T1 through T36). The AF ranges, the number of mutations (#muts), and the mean number of reads supporting the mutations in the AF range (count) are indicated. Created with Biorender.com. (**b**) The mean noise of foreground and recurrent COSMIC mutations in healthy plasma controls (*n* = 46) displayed as kernel densities with individual mutations indicated below. The density distribution of random substitutions is shown in grey. (**c**) Correlation between theoretical AF (calculated from the individual tumors mutation AFs) and the observed AF in sample A of the foreground mutations. Mutations unique to sample A are shown in black (*n* = 59) while mutations reported to be recurrent in COSMIC are shown in blue (*n* = 36). (**d**) AUC for precision–recall and ROC curves at six different AF ranges with error bars indicating the 95%CI on the estimation from 20 Monte Carlo simulations.

**Figure 4 ijms-25-04252-f004:**
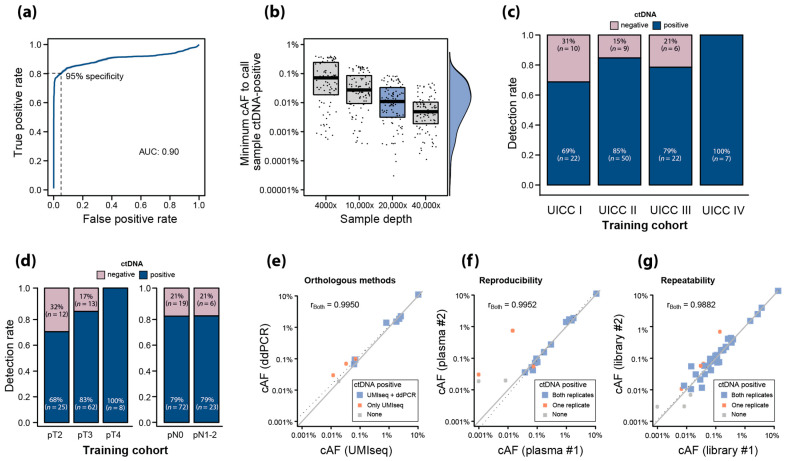
(**a**) ROC analysis on pre-OP plasma samples for UMIseq. Pre-OP plasma samples (*n* = 126) were used as positive labels, and plasma samples from healthy controls against in silico-generated mutational catalogs based on the patient tumors were used as negative labels. The UMIseq sample score threshold for ctDNA detection was set at 95% specificity, as indicated. (**b**) Simulated sample LODs for UMIseq at various sequencing depths (4000×, 10,000×, 20,000×, and 40,000×). The minimum cAF required to call a sample ctDNA-positive was calculated for *n* = 100 in silico-generated mutational tumor catalogs at four sequencing depths. The density (right) indicates the sample LOD distribution at 20,000× mean depth (blue). (**c**) ctDNA detection rates stratified by UICC stage for the training cohort. (**d**) ctDNA detection rates stratified by pT (left) and pN category (right) for the training cohort. (**e**–**g**) UMIseq robustness was evaluated by comparing ctDNA status and cAF levels in paired plasma samples purified from the same blood sample and analyzed with UMIseq and ddPCR (*n* = 11 blood samples) (**e**) or with UMIseq twice (*n* = 19 blood samples) (**f**) and in paired cfDNA samples analyzed with UMIseq (*n* = 46 plasma samples) (**g**). Sample pairs where both, one, or none of the paired samples were called ctDNA-positive are shown in blue, red, and grey, respectively. Using pairs with both samples being ctDNA-positive, the Pearson’s correlation coefficient (r_Both_) and the linear regression line (dotted line) were made. The diagonal is indicated (grey line).

**Figure 5 ijms-25-04252-f005:**
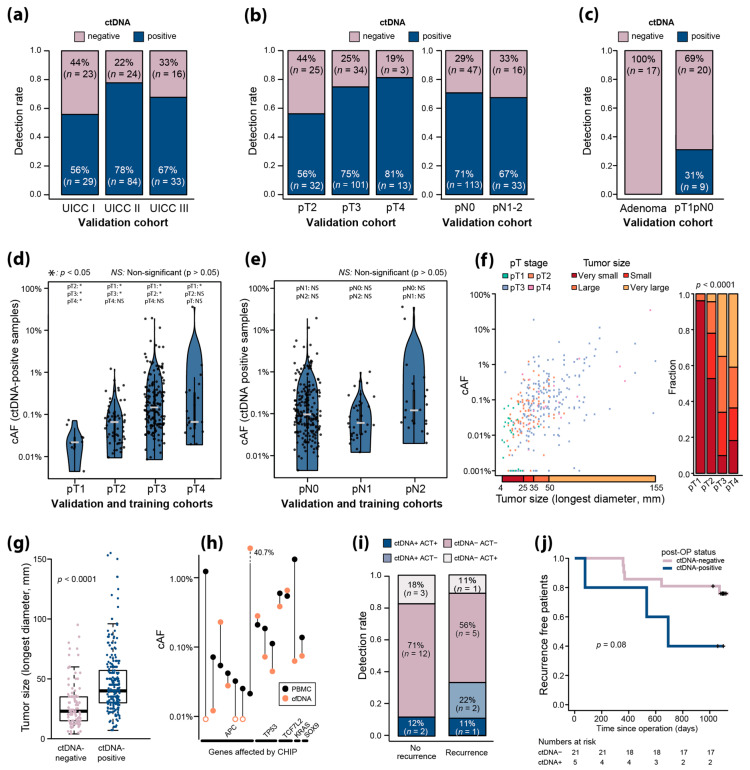
(**a**) ctDNA detection rates stratified by UICC stage for the validation cohort. (**b**) ctDNA detection rates stratified by pT stage (left), and pN stage (right) for the validation cohort. Note that one pT1N2M0 sample (ctDNA-negative) is omitted from the left panel. (**c**) ctDNA detection rates in non-invasive colorectal adenomas and minimally invasive pT1N0M0 tumors. (**d**,**e**) Estimated cAFs of all ctDNA-positive samples in the training and validation cohort (*n* = 223), stratified by pT (**d**) and pN (**e**) categories. Asterisks indicate significant differences (*p* < 0.05, Wilcoxon rank sum test) to other categories as annotated. Grey horizontal lines indicate median cAF. NS = non-significant (*p* > 0.05, Wilcoxon rank sum test) (**f**) Left panel: Correlation between tumor size and cAF. The analysis includes both ctDNA-positive and ctDNA-negative samples (*n* = 130). Twenty-six samples with undetected ctDNA signal (cAF = 0) are displayed with cAF = 0.001%. Right panel: The distribution among tumor sizes of each pT category, displayed as fractions. The quantiles of tumor sizes were used to designate tumors as very small, small, large, and very large, as indicated in the left panel. The *p*-value from a Fisher’s exact test for the association between pT and size is shown. (**g**) Tumor size stratified by ctDNA status. Boxplot indicates median tumor size, 25th and 75th percentile and minimum and maximum value, excluding outliers. Each dot represents a tumor. (**h**) CHIP mutations were identified at 14 positions in 13 patients through UMIseq of PBMCs. cAF of the mutation in the PBMCs and in the cfDNA is shown. Open circles indicate cAF = 0. (**i**) ctDNA detection rates in post-OP samples from recurrent and non-recurrent stage III CRC patients. Samples positive (ctDNA+) and negative (ctDNA-) for ctDNA are shown in grey and blue, respectively, and samples from patients that received adjuvant therapy (ACT+) indicated with darker colors. (**j**) Kaplan–Meier plot of the recurrence free survival time after operation in patients with ctDNA-positive and ctDNA-negative post-OP samples. Patients were censored (indicated with crosses) after the last negative CT scan (30 months or more after operation).

**Table 1 ijms-25-04252-t001:** Cohort characteristics.

	Training Cohort		Validation Cohort		
Characteristic	Stage I–IV CRC	Healthy Controls	Stage I–III CRC	Adenomas and Early CRC	Healthy Controls
*n*	126	37	209	46	24
Gender ^1^					
Female	48 (38%)	18 (49%)	88 (42%)	20 (43%)	9 (38%)
Male	78 (62%)	19 (51%)	121 (58%)	26 (57%)	15 (62%)
Age ^2^	71 (64, 78)	67 (60, 71)	71 (63, 76)	65 (59, 73)	52 (48, 58)
UICC stage ^1^					
Adenoma	-	-	-	17 (37%)	-
I	32 (25%)	-	52 (25%)	29 (63%)	-
II	59 (47%)	-	108 (52%)	-	-
III	28 (22%)	-	49 (23%)	-	-
IV	7 (6%)	-	-	-	-
pT stage ^1^					
Adenoma	-	-	-	17 (37%)	-
pT1	-	-	1 (0%)	29 (63%)	-
pT2	37 (29%)	-	57 (27%)	-	-
pT3	75 (60%)	-	135 (65%)	-	-
pT4	8 (6%)	-	16 (8%)	-	-
Unknown	6 (5%)	-	-	-	-
pN stage ^1^					
Adenoma	-	-	-	17 (37%)	-
pN0	91 (72%)	-	160 (77%)	29 (63%)	-
pN1	18 (14%)	-	32 (15%)	-	-
pN2	11 (9%)	-	17 (8%)	-	-
Unknown	6 (5%)	-	-	-	-
pM stage ^1^					
Adenoma	-	-	-	17 (37%)	-
pM0	120 (95%)	-	209 (100%)	29 (63%)	-
pM1	6 (5%)	-	-	-	-
Synchronous tumors ^1^	-	-	12 (6%)		
Mutations within capture panel ^2^	3 (2, 4)	-	3 (2, 4)	2 (2, 3)	-

^1^ n (%); ^2^ median (2nd quartile, 3rd quartile).

## Data Availability

The sequencing data generated during the study is, due to privacy restriction, available through controlled access from GenomeDK (https://genome.au.dk/library/GDK000009, accessed on 9 April 2024). All other data and programming code supporting the findings of this study are available upon request to the corresponding authors.

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
