# Peer review of "Error-Corrected Deep Targeted Sequencing of Circulating Cell-Free DNA from Colorectal Cancer Patients for Sensitive Detection of Circulating Tumor DNA"

_ijms, 2024, doi:10.3390/ijms25084252_

Round 1

Reviewer 1 Report

Comments and Suggestions for Authors

In this paper, the authors present a technologically advanced assay termed UMIseq, to detect circulating tumor DNA (ctDNA) and further enhancing its role as a potential and promising biomarker. Sequencing-based detection of ctDNA at low tumor fractions is challenging due to the crude error rate of sequencing. They have developed UMIseq, or ultra-deep mutation-integrated sequencing (UMIseq), a fixed-panel deep-targeted sequencing approach. UMIseq features several advantages, including UMI-mediated error correction, exclusion of mutations related to clonal hematopoiesis, a panel of normal samples for error modeling, and signal integration from single-nucleotide variations, insertions, deletions, and phased mutations. The assay was trained and independently validated on pre-operative (pre-OP) plasma from CRC patients (n=364) and healthy individuals (n=61). UMIseq displayed an area under the curve surpassing 0.95 for allele frequencies (AF) down to 0.05%. In the training cohort, the pre-OP detection rate reached 80% at 95% specificity, while it was 70% in the validation cohort. UMIseq enabled the detection of AFs down to 0.004%. To assess the potential for detection of residual disease, 26 post-operative plasma samples from stage III CRC patients were analyzed. In the latter they found that the detection of ctDNA was associated with recurrence. They conclude that UMIseq has a robust performance with high sensitivity and specificity, enabling the detection of ctDNA at low allele frequencies.

Overall, the authors have a done and displayed a thorough evaluation and description of the assay. The AUC close to 1 in the pre op samples is impressive.

I would like them to discuss a few shortcomings

1.     Why was the validation set specificity only 70% in the validation cohort, while it was 95% in the training cohort? And how would that impact the confidence in the assays’ performance?

2.     While do discuss the sensitivity and specificity of the assay in the post-operative setting, as 33 and 89%, respectively, the important outcome is the predictive value (PV), which appears to be 33% for positive PV and 71% for negative PV.

3.     Finally, the use of the assay has 2 potentials, either as screening or as a predictor of response, and the low sensitivity and predictive values, will make it challenging. It would halp if they put this into context with currently available assays, the 2 most common in CRC being the Guardant reveal and Signatera.

Author Response

Response to reviewer's comments are given in red

In this paper, the authors present a technologically advanced assay termed UMIseq, to detect circulating tumor DNA (ctDNA) and further enhancing its role as a potential and promising biomarker. Sequencing-based detection of ctDNA at low tumor fractions is challenging due to the crude error rate of sequencing. They have developed UMIseq, or ultra-deep mutation-integrated sequencing (UMIseq), a fixed-panel deep-targeted sequencing approach. UMIseq features several advantages, including UMI-mediated error correction, exclusion of mutations related to clonal hematopoiesis, a panel of normal samples for error modeling, and signal integration from single-nucleotide variations, insertions, deletions, and phased mutations. The assay was trained and independently validated on pre-operative (pre-OP) plasma from CRC patients (n=364) and healthy individuals (n=61). UMIseq displayed an area under the curve surpassing 0.95 for allele frequencies (AF) down to 0.05%. In the training cohort, the pre-OP detection rate reached 80% at 95% specificity, while it was 70% in the validation cohort. UMIseq enabled the detection of AFs down to 0.004%. To assess the potential for detection of residual disease, 26 post-operative plasma samples from stage III CRC patients were analyzed. In the latter they found that the detection of ctDNA was associated with recurrence. They conclude that UMIseq has a robust performance with high sensitivity and specificity, enabling the detection of ctDNA at low allele frequencies.

Overall, the authors have a done and displayed a thorough evaluation and description of the assay. The AUC close to 1 in the pre op samples is impressive.

We thank the reviewer for the positive feedback and the points raised to improve the manuscript.

I would like them to discuss a few shortcomings

  1. Why was the validation set specificity only 70% in the validation cohort, while it was 95% in the training cohort? And how would that impact the confidence in the assays’ performance?

The specificity of the training cohort was (set at) 95%, as mentioned in line 227, while the specificity of the validation cohort was similar as mentioned in line 283—284. In addition, this is shown in Figure S4, appendix A. The sensitivity, on the other hand, drops from 80% (line 231) in the training cohort to 70% (line 287) in the validation cohort.
To avoid misinterpretation, we have changed the wording in line 231-232 from “UMIseq detected ctDNA in 80% (101/126) of the pre-OP samples” to “Overall, the sensitivity of UMIseq was 80% (101/126) among the pre-OP samples”.

  1. While do discuss the sensitivity and specificity of the assay in the post-operative setting, as 33 and 89%, respectively, the important outcome is the predictive value (PV), which appears to be 33% for positive PV and 71% for negative PV.

We agree that these are important metrics. We have included the PPV and NPV in the main text (line 372-373). The counts for the post-OP samples are in the following categories: TN: 15, FN: 6, FP: 2, and TP: 3. This yields a PPV of 60% (3/5) and NPV of 71% (15/21).

  1. Finally, the use of the assay has 2 potentials, either as screening or as a predictor of response, and the low sensitivity and predictive values, will make it challenging. It would halp if they put this into context with currently available assays, the 2 most common in CRC being the Guardant reveal and Signatera.

We respectfully disagree that the UMIseq assay has the most potential as a screening tool or predictor of response. As mentioned in line 451, UMIseq is developed to detect minimal residual disease (recurrence), as stated in line 447.
To be used as a screening tool, a tumor-agnostic test is required, such as the assay from Guardant. Currently, UMIseq is tumor-informed and thus not suitable as a screening tool. We are aware that the Guardant assay has been used for the detection of minimal residual disease (https://doi.org/10.1158/1078-0432.CCR-21-0410). However, in that study they assess minimal residual disease after the end of definitive treatment (surgery or surgery + adjuvant chemotherapy), while we used plasma samples collected immediately after surgery (before potential adjuvant chemotherapy). Based on this, we argue that a comparison between Guardant and UMIseq inappropriate.

The Signatera assay is tumor-informed. In the JAMA oncology paper from Reinert et al (doi: 10.1001/jamaoncol.2019.0528), the post-OP metrics are as follows: sensitivity: 41%, specificity: 96%, PPV: 70%, NPV: 88% - somewhat higher than for the UMIseq assay. However, we argue that a direct comparison to the Signatera assay is challenging in the post-OP setting due to differences in cohort size and clinical stages.
In our view, a comparison of detection rates (sensitivity) in the larger pre-OP cohorts are more suitable. We have added a paragraph in the discussion section (line 406-409).

Reviewer 2 Report

Comments and Suggestions for Authors

The manuscript “Error-corrected deep targeted sequencing of circulating cell-free DNA from colorectal cancer patients for sensitive detection of circulating tumor DNA” deals with an important issue in the clinical practice related to CRC, that is the development of a method for sensitive and reliable detection of cancer by a minimally invasive procedure, especially in early stages of disease. The topic is interesting to the general readership and it is within the scope of the journal, while the results may have further implications for the improvements in CRC detection and prediction of the biological behaviour of the tumor.  The manuscript is very well structured and well written, with a clear and informative presentation of the scientific background and the potential improvements offered by the proposed sequencing approach. Methodology and the results are described in detail, while the conclusions are well supported by the findings.  There are merely some minor corrections that should be made:

- Tables should be able to stand alone. Even though the authors provided detailed explanations for the Figures, some clarifications from the descriptive part of the Results section should be included in Figure legends. For Figure 5, it should be stated to which samples (which phase of the study) the results correspond to (training, validation, both...). I would suggest that the authors separate this Figure into two, in order to avoid confusion about the group of samples for which the images were generated – images that correspond to the validation group should be separated from others, so that they could easily be compared to results in Table 4.  

- Abbreviations CHIP and LOD should be defined at first usage, not in Material and Methods section that is positioned at the end of the manuscript.

Author Response

Response to reviewer comments in red:

The manuscript “Error-corrected deep targeted sequencing of circulating cell-free DNA from colorectal cancer patients for sensitive detection of circulating tumor DNA” deals with an important issue in the clinical practice related to CRC, that is the development of a method for sensitive and reliable detection of cancer by a minimally invasive procedure, especially in early stages of disease. The topic is interesting to the general readership and it is within the scope of the journal, while the results may have further implications for the improvements in CRC detection and prediction of the biological behaviour of the tumor.  The manuscript is very well structured and well written, with a clear and informative presentation of the scientific background and the potential improvements offered by the proposed sequencing approach. Methodology and the results are described in detail, while the conclusions are well supported by the findings.  There are merely some minor corrections that should be made:

We thank the reviewer for the positive feedback and the points raised to improve the manuscript.

- Tables should be able to stand alone. Even though the authors provided detailed explanations for the Figures, some clarifications from the descriptive part of the Results section should be included in Figure legends. For Figure 5, it should be stated to which samples (which phase of the study) the results correspond to (training, validation, both...). I would suggest that the authors separate this Figure into two, in order to avoid confusion about the group of samples for which the images were generated – images that correspond to the validation group should be separated from others, so that they could easily be compared to results in Table 4.  

We agree with the reviewer on this point. We have not split figure 5 into two figures. Instead, we have clearly stated whether the samples are part of the training or validation cohort by adding this to the figure legends (lines 247-249 and lines 298-300). Furthermore, we have updated figure 4c, 4d, and 5a-e with a label indicating the training/validation cohort.

- Abbreviations CHIP and LOD should be defined at first usage, not in Material and Methods section that is positioned at the end of the manuscript.

This has been updated. Line 110-111 for CHIP. We have moved the LOD abbreviation from the Figure 2 legend (line 148) to the main text (line 158).